# MetaTiME integrates single-cell gene expression to characterize the meta-components of the tumor immune microenvironment

Yi Zhang [1,2], Guanjue Xiang [1,2], Alva Yijia Jiang[1], Allen Lynch[1,2], Zexian Zeng[1,2], Chenfei Wang [1,2], Wubing Zhang[1,2], Jingyu Fan [1,2], Jiajinlong Kang[1], Shengqing Stan Gu[3], Changxin Wan[1,2], Boning Zhang[1,2], X. Shirley Liu[1,2,4] ✉, Myles Brown [3,4] ✉ & Clifford A. Meyer [1,2,4] ✉

Recent advances in single-cell RNA sequencing have shown heterogeneous cell types and gene expression states in the non-cancerous cells in tumors. The integration of multiple scRNA-seq datasets across tumors can indicate common cell types and states in the tumor microenvironment (TME). We develop a data driven framework, MetaTiME, to overcome the limitations in resolution and consistency that result from manual labelling using known gene markers. Using millions of TME single cells, MetaTiME learns meta-components that encode independent components of gene expression observed across cancer types. The meta-components are biologically interpretable as cell types, cell states, and signaling activities. By projecting onto the MetaTiME space, we provide a tool to annotate cell states and signature continuums for TME scRNA-seq data. Leveraging epigenetics data, MetaTiME reveals critical transcriptional regulators for the cell states. Overall, MetaTiME learns data-driven meta-components that depict cellular states and gene regulators for tumor immunity and cancer immunotherapy.

Recent advances in cancer research have implicated the integral function of the tumor microenvironment (TME) in tumor progression and therapy responses[1–6]. Understanding interactions between cancer cells and the non-cancer compartments, including immune cells, fibroblasts, and endothelial cells, has shown potential targets for cancer immunotherapy. Specifically, single-cell RNA-sequencing (scRNA-Seq) applied on multiple patient tumors has enabled the high-resolution identification of TME constituents that interfere with the elimination of cancer cells. For example, exhausted tumor-infiltrating lymphocytes (TILs)[4,7,8], and certain tumor-associated macrophages subtypes[9–11], have been associated with tumor development. However,

the definition of cell types and cell states in tumor scRNA analyses still relies on manual labeling by experts using known exclusive biomarkers following unsupervised clustering[12,13], which lacks consistency and varies between different cohorts.

As single-cell data accumulate, integrating a large collection of cells from multiple cohorts can help unify the definition of cell types and states to facilitate the automatic annotation of new scRNA-seq data[14,15]. One approach to cell annotation is to use predefined biomarker lists. However, these biomarkers might not cover domain-specific cellular states, for example, well-defined immune cell markers derived from blood immune cells may not fully cover the TME disease

[1]Department of Data Science, Dana-Farber Cancer Institute, Boston, MA 02215, USA. [2]Department of Biostatistics, Harvard T.H. Chan School of Public Health, Boston, MA 02215, USA. [3]Department of Medical Oncology, Dana-Farber Cancer Institute, Boston, MA 02215, USA. [4]Center for Functional Cancer Epigenetics, Dana-Farber Cancer Institute, Boston, MA, USA. ✉e-mail: xsliu.res@gmail.com; myles_brown@dfci.harvard.edu; cliff_meyer@ds.dfci.harvard.edu

context[14]. Moreover, although cell type definitions in reference databases such as CIBERSORT, Azimuth, and Human Primary Cell Atlas[16–18] can be useful, the granularity of these definitions varies between databases. Several efforts integrating pan-cancer scRNA data have revealed subtypes in the TME through the manual annotation of clusters using a shortlist of exclusive gene markers[13,19,20].

Another approach is to map a dataset containing unannotated cell states onto an annotated reference. Methods to obtain such representations include canonical correlation analysis (CCA)[21], adjusted principal components (Harmony)[22], or generative deep learning models using variational autoencoders (scVI)[23]. These methods use dimension reduction onto a common latent space to align cells with similar states between datasets, without ascribing meaning to the latent space representations. An alternative data driven approach is to identify low dimensional latent space representations in which a biological meaning can be ascribed to each latent dimension. Several matrix factorization algorithms have been developed to represent high dimensional data in a low-dimensional space with interpretable components, including non-negative matrix factorization (NMF)[24] and independent component analysis (ICA)[25,26].

Here we develop a computational framework for mapping millions of single cells from multiple cohorts onto a comprehensive and interpretable latent space, learnt from the data. The framework, MetaTiME (Meta-components of the Tumor immune MicroEnvironment), identifies reproducible low-dimensional meta-components that reflect independent components of gene expression variation across cohorts and cancer types. MetaTiME adopts ICA for dimensional reduction to maximize the mutual independence among components. We use MetaTiME to obtain meta-components (MeCs) from 1.7 million single cells across 79 tumor datasets. These MeCs represent the TME landscape along 86 data-driven transcriptional directions mirroring lineage-specific cell states and signaling activities. Furthermore, we develop a MetaTiME toolkit for using the MeCs to annotate cellular states and signature continuums in tumor scRNA datasets, and to reveal differential signatures across immunotherapy responses. Finally, by incorporating transcription factor binding data, MetaTiME identifies and prioritizes putative transcriptional regulators that may modulate tumor immunity.

## Results

### MetaTiME as a general framework to discover consensus transcriptomic programs

The MetaTiME framework consists of three stages: meta-component (MeC) discovery, interpretation of MeCs, and application of cell state annotations (Fig. 1a). The MeC discovery stage detects repeatable sources of variation from multiple single-cell measurements sharing similar cellular properties. The MeC interpretation step involves a one-time curation effort using biomarker databases, pathway information and Cistrome DB chromatin profiling data[27]. In the third step, users map MeCs onto their new tumor scRNA-seq datasets using MetaTiME application tools, to obtain annotated cell states and signature continuums.

To train MeCs for the TME context, we collected and curated 2,157,387 cells from 76 studies ranging across 27 cancer types, using publicly available tumor scRNA-Seq data mostly from TISCH[28]. After removing the TISCH annotated malignant cells using MAESTRO[17], 102,703 stromal cells and 1,617,110 immune cells were retained for downstream training (Supplementary Fig. 1, Supplementary Data 1).

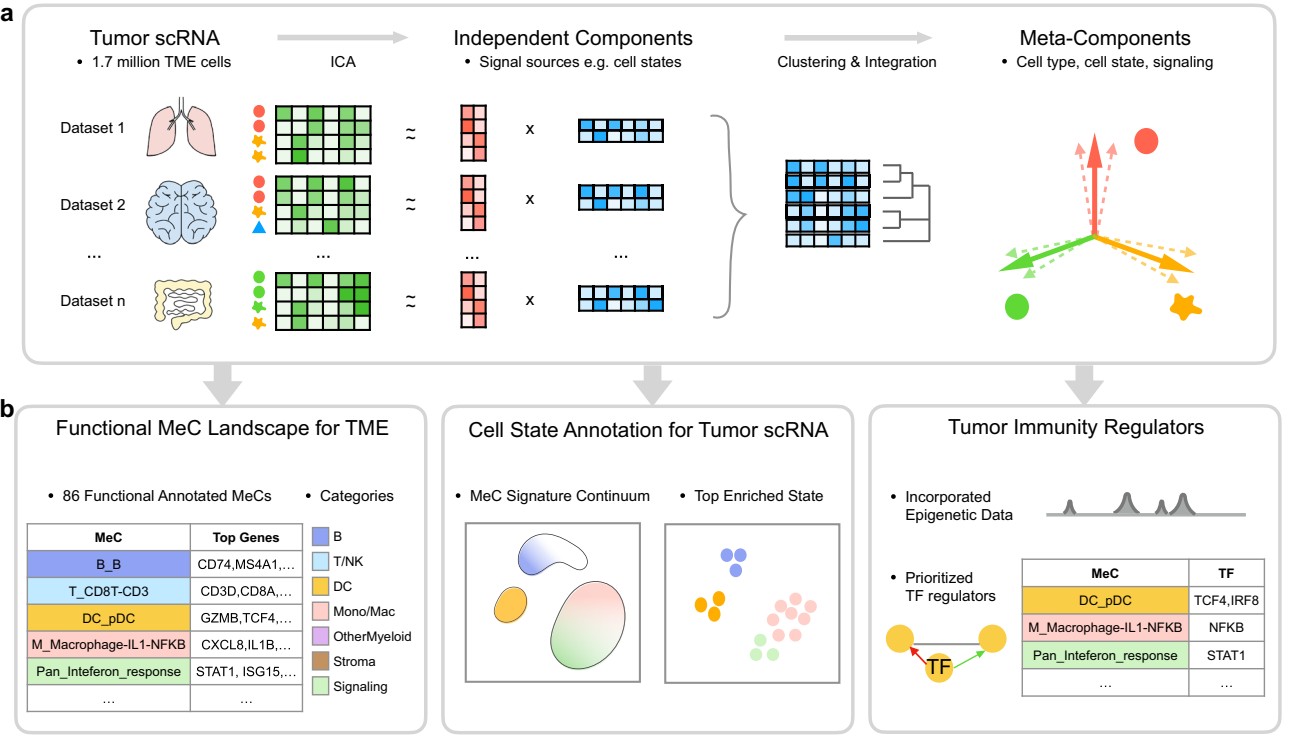

**Fig. 1 | Overview of MetaTiME.** MetaTiME integrates 1.7 million single cells to learn common transcriptional programs in the tumor microenvironment (TME). **a** Steps for Meta-components (MeCs) discovery. For each scRNA dataset, the expression matrix of TME cells is decomposed into a loading matrix (red) and an independent component (IC) matrix through independent component analysis (ICA). The ICs represent mutually independent sources of transcriptional variation. ICs from each dataset are concatenated and clustered into groups of ICs with high similarity, representing transcriptional programs shared across TME. MeCs are then calculated as averaged profiles of ICs from each cluster. Each MeC is interpretable, representing gene signatures of cell type, cell states, or signaling pathway activities. **b** Left: MetaTiME provides 86 functionally annotated MeCs that depict the TME transcriptional landscape. They are grouped into six lineage-related categories and one category reflecting signaling activities, each using a background color corresponding to the lineage. Middle: the MetaTiME annotation tool facilitates automatic annotation of cell states for new tumor scRNA data. Right: candidate regulators of each MeC are prioritized by combining MeC gene weights with epigenetics data. MeC: meta-components, TME: tumor microenvironment, ICA: independent component analysis, MeC: meta-component, TF: transcription factor.

The 76 studies were further partitioned according to cancer type, resulting in 93 datasets, including 7 datasets with immune checkpoint blockade (ICB) treatment and 3 10x Genomics provided datasets representing peripheral blood mononuclear cells (PBMC) sampled from healthy donors.

In the MeC discovery stage, MetaTiME first decomposes the log-transformed expression matrix of each single dataset using Independent Component Analysis (ICA)[25]. We adopted ICA to maximize mutual independence among gene expression components. In simulations ICA performed slightly better than Non-negative Matrix Factorization (NMF) in simulated single-cell data with pre-embedded transcriptional signatures (Supplementary Fig. 2). The feature weight distribution of each Independent Component (IC) also enables normalization of the gene contribution scores for measuring similarity among components. MetaTiME then applies two transformations to the IC vectors, $z$-weight normalization and skewness alignment, to ensure the scales of gene representation scores are comparable among components (**Methods**, Supplementary Fig. 3a). Next, MetaTiME filters ICs to retain ones that are reproducible across multiple cohorts (the minimum Pearson correlation with any other IC ≥3). These are passed to the Louvain graph clustering algorithm to merge IC groups into MeCs (**Methods**). Lastly, MetaTiME computes averaged profiles of gene $z$-weights within each IC cluster, yielding 86 MeCs trained for the TME (Fig. 1a, Fig. 2a). The number of MeCs was automatically determined by simultaneously optimizing granularity and independence in IC clustering (Supplementary Fig. 3b). Importantly, the MeC clustering does not depend on cohort source (Supplementary Fig. 4a, cohort source of component), and is robust during leave-one-out testing in which the MeCs are learnt with a single cohort left out of the analysis (Supplementary Fig. 5). The leave-one-out MeCs are highly correlated with the MeCs learnt on the complete data (mean maximum correlation = 0.988), and the numbers of components are similar (mean leave-one-out MeC number = 84.64, full-set MeC number = 86). This integration after decomposition approach overcomes batch effects, which are often a challenge in single cell RNA-seq data analysis. Moreover, we implemented the standard approach in which cluster-wise signatures are obtained by batch effect removal, clustering, and differential expression analysis (Supplementary Fig. 6a). Limited by server computing memory, the maximum number of datasets we could integrate in this way was 21, and the cluster-wise signatures display a lower level of specificity in the test data (Supplementary Fig. 6b,c). Therefore, the MetaTiME approach allows for the effective integration of large numbers of single cell datasets, which will become increasingly important as data accumulates.

## MetaTiME defines interpretable meta-components

In principle, each MeC represents one independent source of transcriptional variation commonly present in the TME. We investigated top ranked genes in MeCs and found MeCs are highly interpretable, reflecting common biological processes in the TME. For instance, the MeC derived from the largest IC cluster is highly enriched in interferon response genes, such as *ISG15*, *IFI6*, *LY6E*, and *MX1*, indicating that the underlying interferon response is among the most common source of transcriptional variation shared across tumor samples and cohorts (Fig. 2a, b). Intriguingly, top genes of each MeC are enriched in known biomarkers or regulators. For example, several T cell-related MeCs identify different gene modules co-expressed in T cells reflecting activation of different T-cell related processes (Fig. 2c, Supplementary Fig. 7, Source Data 1). MeC-65, T cell co-signaling, features T cell receptors in co-stimulatory and co-inhibitory pathways[29,30], such as *TNFRSF4* (OX40), *TNFRSF18* (GITR) *TNFRSF9* (4-1BB), and *ICOS* (Fig. 2c, left). MeC-40, CXCL13 + exhausted CD8 T cell, features receptors characterizing the exhausted CD8 T cell state[8], including *HAVCR2* (TIM3), *LAG3*, *TIGIT*, and *PDCD1* (PD1) (Supplementary Fig. 7, Source Data 2), each being potential ICB targets[31]. In addition, this MeC is

characterized by a high level of *CXCL13* (Fig. 2c, second panel), a cytokine mediating immune cell trafficking to tertiary lymphoid structures[32]. In contrast, a related MeC representing T cell co-signaling receptors in regulatory CD4T cells (Treg) has a different ranking, including *TNFRSF18*, *TNFRSF4*, *TIGIT*, *TNFRSF1B*, *CTLA4*, *CD27* among the top 20 genes, along with the regulatory T cell-specific marker *FOXP3* (Fig. 2c, right, Source Data 1). Though ICB has been an extremely successful therapy for some patients, it has not yet had an impact on the majority of patients[33]. Investigating the top members in the MeCs involving T cell receptor pathways may help identify new ICB targets.

## MetaTiME depicts the functional landscape of transcriptomic variation and cell states in the tumor microenvironment

We provided functional annotations of all MeCs by examining top $z$-weight genes and compared these with functional gene sets, such as immune cell type markers[15,18] and gene ontology databases[34]. We found that 86 MeCs clearly mirror gene expression patterns corresponding to cell types, cell states and signaling pathway activities, depicting a landscape of non-cancer cell states in the TME (Fig. 2, Supplementary Data 2: MeC annotation). The top genes of the cell type MeCs match well-known lineage-specific markers[15,18]. Examples include *CD74*, *CD79A*, *MS4A1* for B cells (MeC-18, B cell), *CD3D*, *CD8A*, *CD8B* for T cells (MeC-32, *CD3* + CD8 T cell), and *LYZ*, *VCAN*, *S100A9* for CD14 + Monocytes (MeC-17, CD14 monocyte) (Fig. 2b, Fig. 2d, Source Data 3). The majority of MeCs define high resolution lineage-specific cell states (Fig. 2b, Supplementary Fig. 4b). Taking the B cell lineage as an example, multiple MeCs harbor genes specific to B cell developmental stages[35], ranging from a progenitor B cell state (*CD69* and *PAX5* in MeC-50, *PAX5* B cell), to a mature B cell state (*CD79A* in MeC-18, B cell), an antibody-secreting plasma cell state (*XBP1* in MeC-4, plasma B cell; *JCHAIN* in MeC-77, *JCHAIN* + plasma B cell), and immunoglobulin secretion states (IGK and IGH in MeC-30, immunoglobin kappa B cell; IGL and IGH in MeC-50, immunoglobin lambda B cell) (Fig. 2b). Lastly, like the interferon responsive MeC mentioned above, we found a subset of MeCs that are more accurately interpreted as signaling pathways because their top genes are more related to pathways or molecular functions than to cell identities.

We organized the 86 annotated MeCs into six cell lineage-focused categories and one signaling pathway-focused category (Fig. 2a,b and Supplementary Data 2: MeC annotation, MeC enrichment). Among these, there are 6 B lineage-related MeCs for B cells; 20 T cell lineage MeCs covering CD8 T cells, CD4 T cells, and natural killer (NK) cells; 4 dendritic cell (DC) lineage MeCs; 12 monocyte and macrophage-related MeCs; 3 platelet, erythrocyte, and mast cell MeCs; 6 stromal cell-related MeCs for fibroblasts, myofibroblasts and endothelial cells; and 35 MeCs in the signaling category (Fig. 2b). We demonstrated that the MeCs are of high specificity, visualizing the log-scaled $z$-weights of known cell subtype markers and pathway biomarkers (Fig. 2d). Correlating MeCs with the comprehensive immune cell type database Azimuth[15] validated the lineage-specificity of several MeCs, while most MeCs reflect cell states that appear specific to the tumor context (Supplementary Fig. 4b).

## MetaTiME annotates cell states and signature continuums when applied to the tumor microenvironment single-cell data

As MetaTiME MeCs provides a highly interpretable basis for the TME in single cells, we provided a toolkit to discover MeC signature continuums and enriched cell states in scRNA-seq TME data (code deposited in https://github.com/yi-zhang/MetaTiME). The MetaTiME annotation toolkit takes as input the scRNA-seq expression matrix after depth normalization and log transformation, maps each single cell onto the pre-trained MeC space, and annotates the most highly enriched cell states for pre-defined cell clusters. The cell clusters are by default calculated using graph clustering with high resolution after an

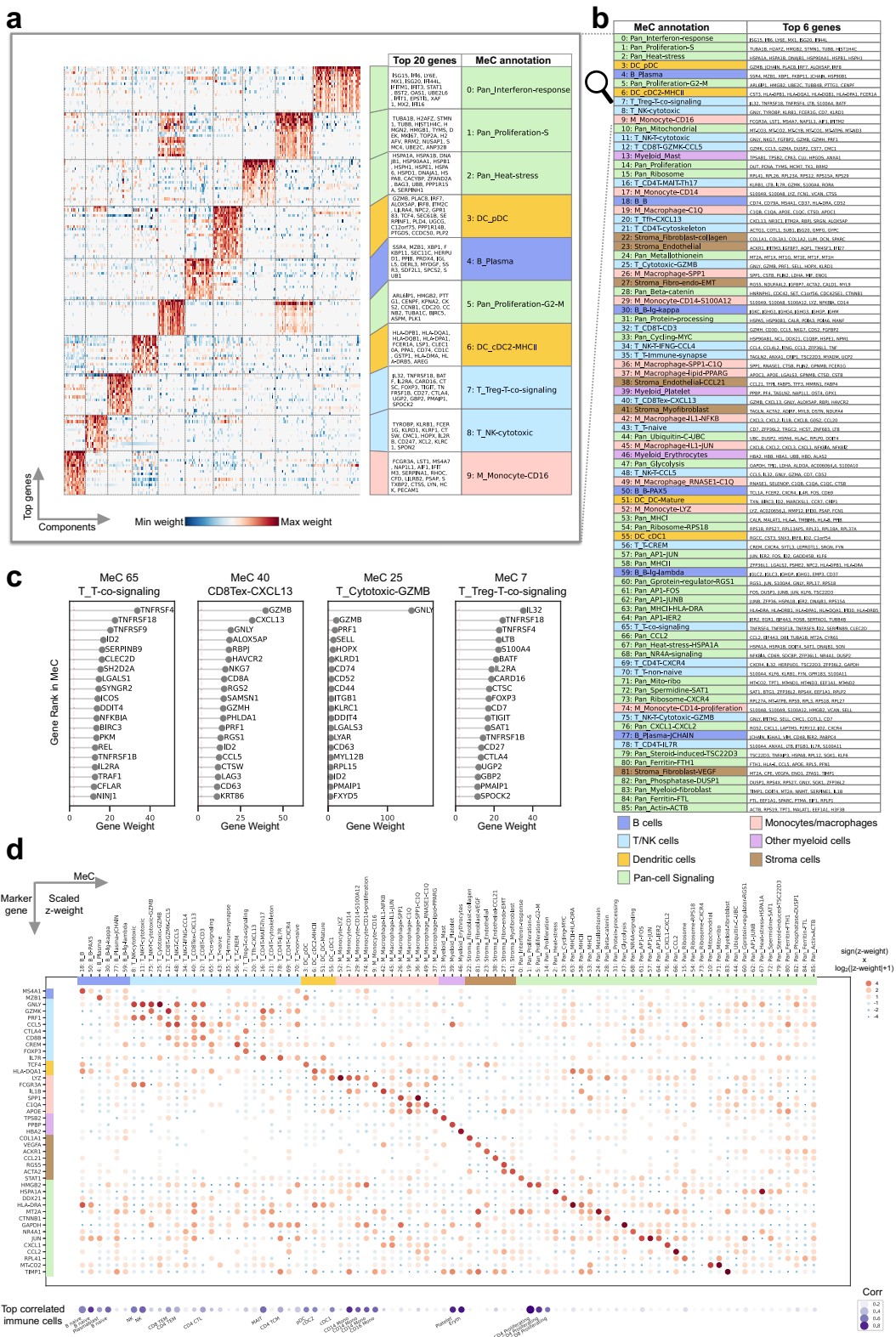

**Fig. 2 | MetaTiME meta-components are biologically interpretable with top genes.** **a** Heatmap of top ten most recurrent clusters of MeCs showing normalized gene weights. **b** Biological characterization of each MeC with top genes. To facilitate biological interpretation, MeCs are categorized into six lineage-associated classes (B cells, T cells for CD4T, CD8T and NK cells, dendritic cells, monocyte and macrophages, other myeloid cell types, and stroma cells) and one signaling pathway-associated class. **c** Examples of T cell related MeCs with top 20 genes with largest weights. **d** Gene contribution of known lineage-related biomarkers for each

MeC, and correlation with known immune markers from Azimuth. In the top dot plot, size and color represents log-scaled MeC z-weights of each gene in each MeC. In the bottom dot plot, size and color represents the maximum correlation coefficient between MeC and Azimuth defined marker genes per cell type. MeC meta-component, DC dendritic cell, Mono/Mac monocytes and macrophages. Each MeC and gene is shaded with a background color corresponding to the lineage category. Source data are provided as a Source Data file.

optional batch effect correction with Harmony[22]. We demonstrate the application of MetaTiME on basal cell carcinoma (BCC) single-cells from Yost et al.[8]. These test cells were excluded from the MetaTiME training stage. MetaTiME annotates the enriched cell states (Fig. 3a) highlighting gradients of exhausted CD8 T cells and follicular helper T cells (Tfh) (Fig. 3c). The most enriched cell states consistently match the manual labelling from the original study with improved resolution (Fig. 3b) and highly express corresponding markers (Fig. 3d, Source Data 4). In addition, compared to Seurat's[14] automated CIBERSORT marker-based annotations (14 cell types, Supplementary Fig. 6a), MetaTiME provides higher resolution (38 cell states, Fig. 3b). Other automatic annotation gene panels were also tested, including the human primary cell atlas (HPCA) panel and Blueprint-ENCODE panel used in SingleR[16], where macrophages and plasma cells appear to be mislabeled as subclusters within T cell clusters (Supplementary Fig. 8b, c). Interestingly, the MetaTiME annotation not only indicates the CD8 T cell and CD4 T cell subtypes, but also splits cells further into cell states with polarized expression in proliferation, cytotoxicity, exhaustion level, heat stress, co-signaling pathways, and so on. (Fig. 3a, Supplementary Fig. 8a). The B cell group is further partitioned into distinct B cell developmental states including a B cell cluster with cell cycle and MYC activities (Fig. 3a,b, Supplementary Fig. 8), which possibly represent germinal center (GC) B cells undergoing active expansion and maturation[36].

We thus re-annotated all tumor scRNA cohorts using MetaTiME and investigated the distribution of cell state compositions across cancer cohorts. As shown in Fig. 3e and Supplementary Fig. 9, tumors are highly heterogenous and the TME cellular composition is only partially determined by cancer type (Source Data 5–6). For example, Cholangiocarcinoma (CHOL) is highly enriched in stromal cells including collagen-secreting fibroblast, as expected[37], while other samples including ovarian cancer (OV), pancreatic adenocarcinoma (PAAD), and multiple myeloma (MM) are also stromal-rich. Furthermore, tumors with high infiltration of the MeC-12, GZMK + CCL5 + CD8 T cell state, include multiple tumor types including bladder cancer, breast cancer, and skin cancer, suggesting that immune infiltration is sample-dependent and that cancer treatments should be personalized[38].

## Differential MetaTiME analysis detects alterations of transcriptional programs in immunotherapy

Single-cell data derived from ICB trials is invaluable for identifying cell types associated with ICB treatment or response[8]. However, the detection of differential cell type abundances in ICB cohorts has been challenging due to the heterogeneity of cell type proportions and to the limited numbers of patients in each cohort[39]. We compared differences in MeC signatures instead of cell count proportions, to understand immune responses during ICB. We analyzed two ICB cohorts, a basal cell carcinoma (BCC) cohort with samples from pre- or post-ICB treatment[8], and a bladder cancer (BLCA) cohort with samples from ICB responders and non-responders[40]. We applied MetaTiME for per-cluster cell state annotation and per-cell MeC signature evaluation. For each cell state cluster, we tested all MeC signatures passing significance (average z-weight ≥2) between conditions using the two-sided t-test. We plotted cluster-wise signatures in the significance – effect size scatterplot to highlight the most significant differential MeCs (**Methods**). In a comparison of pre- and post-ICB treatment, we observed higher expression of cytotoxic T cell and B cell MeCs in the post-ICB samples. Moreover, several monocyte and macrophage states are also suppressed after ICB treatment (Fig. 4a, Source Data 7). Notably, the IL1B-positive macrophage signature is also found to be elevated in non-responders compared to responders in the BLCA ICB cohort (Fig. 4b, Source Data 8). Since activation of the IL1B pathway is a known regulator of inflammatory processes[41], we sought to investigate whether the IL1B-positive macrophage signature is associated with tumor survival prognosis in bulk RNA-seq data from The Cancer Genome Atlas Program (TCGA). We evaluated TCGA tumors using the averaged expression of the top 20 genes from MeC-42, M_Macrophage-IL1-NFKB, which ranks first in elevated MeCs in non-responders (Fig. 4a). We found that higher expression of the IL1B signature is associated with lower survival rate in multiple cancer types, especially in Low Grade Glioma (LGG) and in Kidney renal cell carcinoma (KIRC) (Supplementary Fig. 10). This suggests the macrophage state with IL1B pathway activation is associated with poor prognosis and lower ICB efficacy.

## MetaTiME delineates myeloid cells in different metabolic states

As specific myeloid cell states have been associated with cancer survival and treatment response, we sought to systematically characterize MeCs related to monocytes and macrophages. Although the canonical definition of M1 and M2 macrophages is derived from cytokine polarized macrophages in vitro[42], MetaTiME's myeloid-related MeCs represent a more complex framework for understanding tumor-infiltrating macrophages. MetaTiME's 12 monocyte and macrophage related MeCs can be summarized into six central monocyte or macrophage states for the TME, after merging similar states such as MeC-42, IL1-NFkB Macrophage and MeC-45, IL1- JUN Macrophage, due to similarity among top genes (Fig. 4c). Monocytes are classified as two categories, CD14 + and CD16 + . For macrophages, four MeCs define common states of intra-tumor macrophages: C1Q + , SPP1 + , lipid-rich, and IL1B + macrophages, and two MeCs, representing interferon and MHC-II signaling pathways, are less frequently observed among macrophages (Fig. 2b, Fig. 4c). In comparison, previous studies defined different tumor associated macrophages in terms of manually selected representative genes after clustering myeloid cells. For example, Cheng et al.[10] defined several tumor associated macrophage types including ISG15 + , SPP1 + , INHBA + , VCAN + , NLRP3 + , and FN1 + macrophages, while Bi et al.[43] defined CXCL10-high, GPNMB-high, FOLR2-high, VSIR-high, and cycling macrophages for advanced renal cell carcinoma (ccRCC). We find that the MetaTiME-defined macrophage MeCs reflect co-expression relationships with the selected marker genes. For example, macrophage markers from Bi et al. rank high in several macrophage-related MeCs (Supplementary Fig. 11), and the expression pattern of the marker genes picked by Bi et al. (CXCL10, GPNMB, VSIR, FOLR2, Cycling marker MKI67) correspond to several MeCs (MeC-0, interferon response; MeC-37, PPARG + lipid-rich macrophage; MeC-58, MHCII-high, and MeC-49, RNASE1 + , C1Q + macrophage) (Supplementary Fig. 11). However, MetaTiME reveals additional distinct components such as the SPP1 + and C1Q + MeCs, which were detected as separate myeloid types in the Cheng et al. multi-cohort study (Supplementary Fig. 12). While the manual reconciliation of cell types from multi-cohort scRNA data shows many marker genes to be consistent with the top genes in the MetaTiME MeCs, the myeloid cell population is not neatly partitioned into cell clusters and might be better represented in terms of expression signature continuums. For example, when mapping myeloid MeCs onto the kidney myeloid cells, the IL1B + MeC signature (MeC-42, MeC-45) is distributed across the M09_Macro_IL1B cluster as well as the CD14 monocyte cluster (Supplementary Fig. 12).

To investigate functional differences among the different macrophage states, we applied gene set enrichment (GSEA)[34] analysis using the top MeC genes (**Methods**). Interestingly, the different macrophage states have different metabolic preferences (Fig. 4d, Source Data 9). Glucose metabolism and the glycosylation pathway are highly active in SPP1 + macrophages, while lysosome and phagosome activity are the most highly enriched in C1Q + macrophages. Lysosome and cholesterol metabolism, including PPARG signaling, are enriched in the lipid-rich state. The inflammatory IL1B and NFkB pathways are highly active in IL1B + macrophages. Several macrophage states are related to cell signaling. SPP1 for example, encodes Osteopontin, which has been

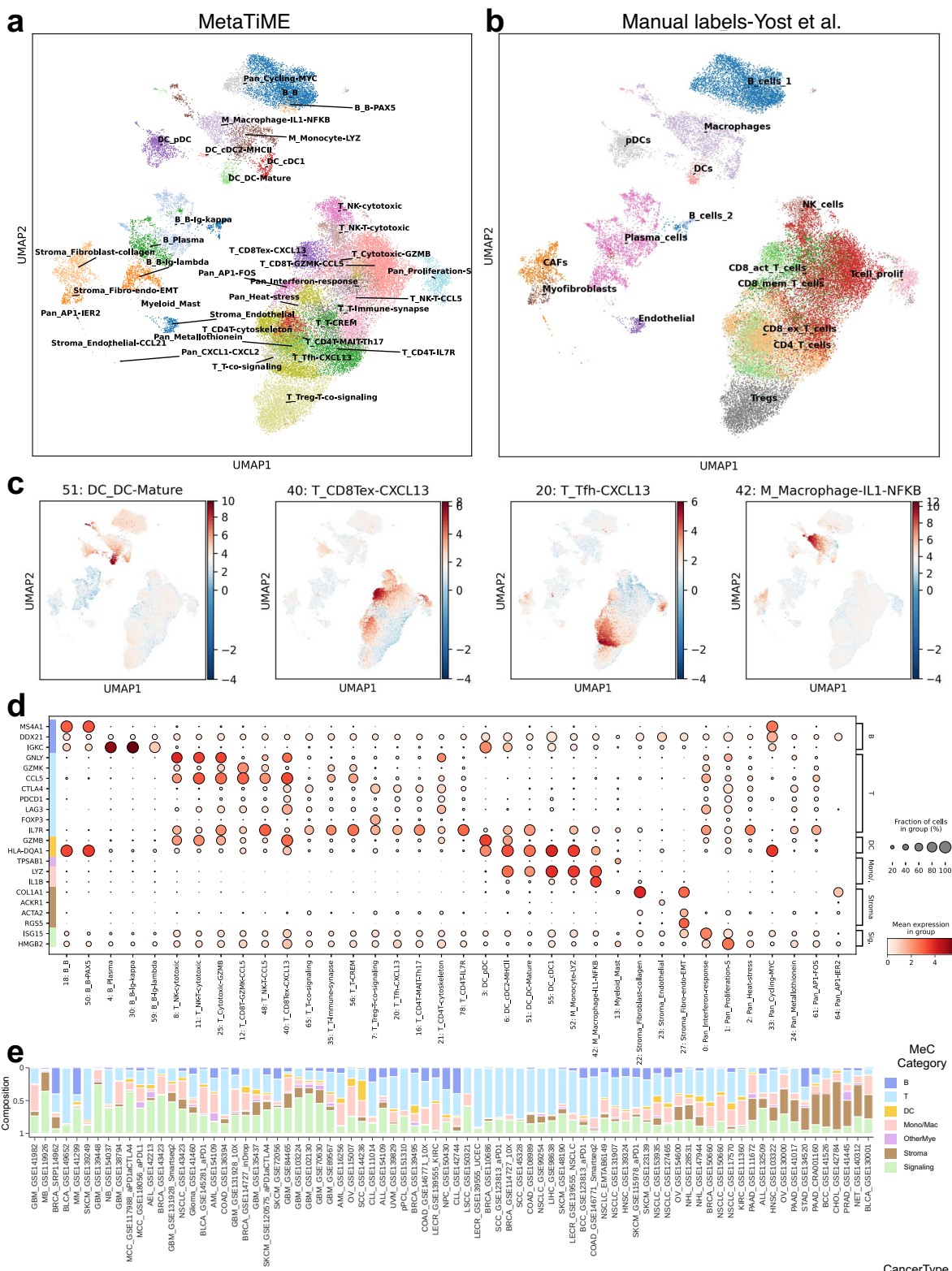

**Fig. 3 | MetaTiME annotates cell states with high resolution on tumor microenvironment single-cell data. a** MetaTiME cell state annotation of cell clusters in a basal cell carcinoma scRNA dataset based on top enriched MeCs. **b** Manual annotation labels by experts from the original study shown on the same UMAP space. **c** Signature continuums of four MeCs representing the mature dendritic cell state, the CXCL13-secreting exhausted T cell state, the CXCL13-secreting T follicular helper cell state, and the IL1B pathway-activated macrophage state. **d** Marker gene expression for each annotated cell cluster as in (**a**). **e** Bar plot showing cell state composition of tumor microenvironment for tumor scRNA dataset cell states. The proportion of cell states from the same MeC category are aggregated. Source data are provided as a Source Data file.

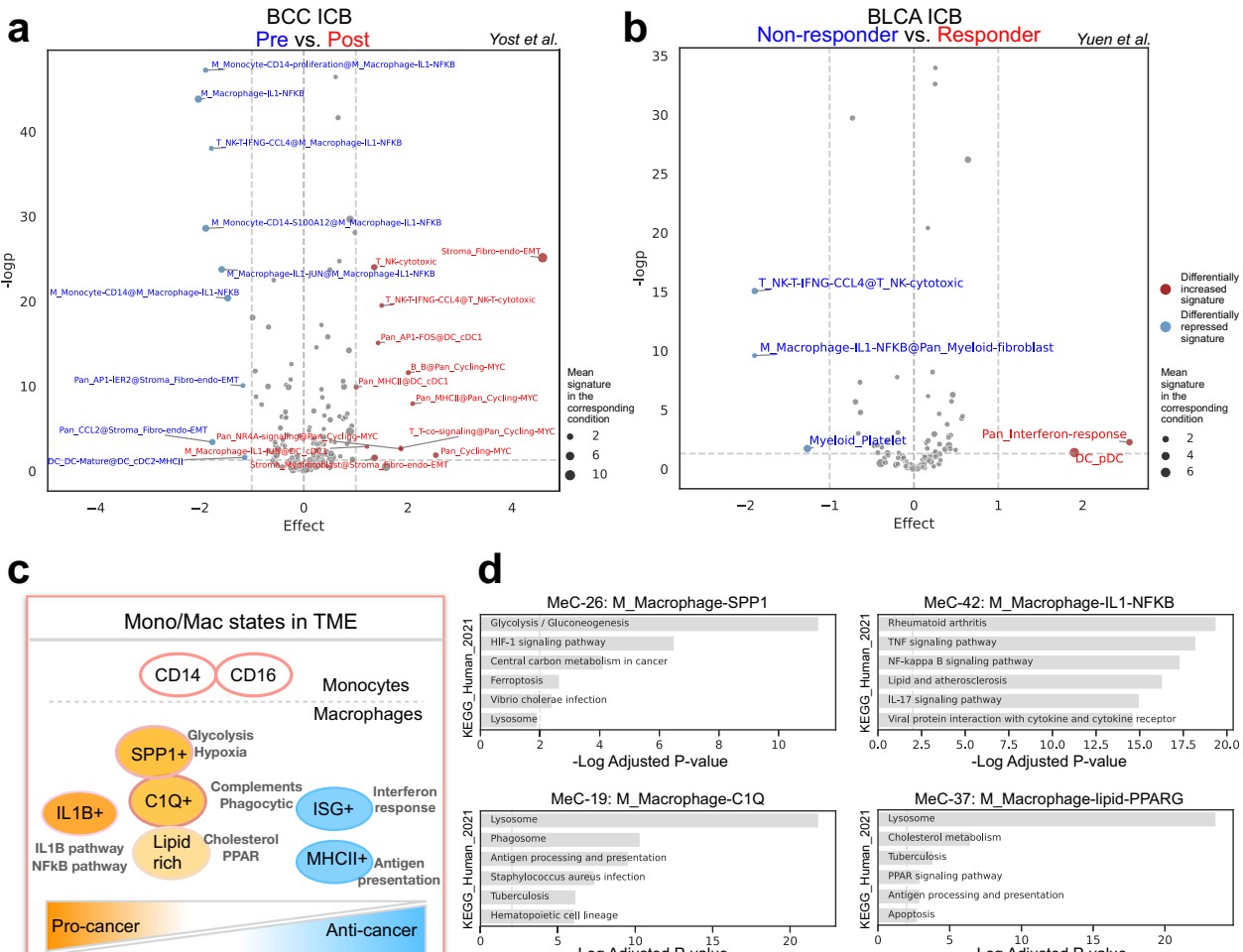

**Fig. 4 | Differential signature analysis and delineated macrophage states in TME. a** Differential MeC signature testing for enriched cell states comparing pre- and post- immunotherapy conditions in basal cell carcinoma (BCC) with two-sided *t*-test. *X*-axis: Difference of mean signature scores between post- and pre-immunotherapy conditions. *Y*-axis: -log(p-value) from two-sided *t*-test. The significant cluster-wise differential signature is marked as "EnrichedMeC@ClusterName"; when the enriched MeC is the same as cluster name, the signature is marked "ClusterName". Red dots, differentially increased signature; size of the dots is proportionally to the mean signature score of cells from the post-immunotherapy condition in the cluster. Blue dots, differentially repressed signature; size of the dots is proportionally to the mean signature score of cells from the pre-immunotherapy condition in the cluster. **b** Differential signature testing for enriched cell states comparing non-responders and responders from pre-treatment condition in bladder carcinoma (BLCA) with two-sided *t*-test. Red dots, differentially increased signature; size of the dots is proportionally to the mean signature score of cells from the responder condition in the cluster. Blue dots, differentially repressed signature; size of the dots is proportionally to the mean signature score of cells from the non-responder condition in the cluster. **c** Model of monocytes and macrophage states in tumor and their metabolic differences. **d** Top pathways enriched in different macrophage MeCs, with adjusted hypergeometric tested *p*-value from Enrichr. BCC basal cell carcinoma, BLCA bladder carcinoma. Source data are provided as a Source Data file.

found to foster an environment that promotes cancer metastasis[44]. The *C1Q* + MeC features *C1QA*, *C1QB*, and *C1QC*, members of the family of complement molecules that could have dual functions in chronic inflammation[45]. The *IL1B* + meta-components features cytokines co-expressed with *IL1B*, including *CXCL8*, *CXCL2*, and *CXCL3*, all of which can interact with other cells in the TME by binding to cytokine receptors[46] (Fig. 2b).

## Incorporation of epigenetic data prioritizes transcriptional regulators of tumor immunity

We next investigated the transcription factors (TFs) that regulate the MeCs, hypothesizing that the co-expression of genes in a subset of MeCs is determined through TF regulatory events. Our group previously developed the Cistrome Data Browser and Lisa to predict transcriptional regulators of gene sets based on chromatin immuno-precipitation with sequencing (ChIP-seq) data[27,47]. Thus, we used Lisa to predict the TFs that regulate the top genes of each MeC, and compared these Lisa regulatory prediction scores with the MeC

*z*-weights across TFs (Supplementary Data 3). We found that, for many MeCs, the same TFs were predicted to be both regulators of the MeC and were highly expressed in the MeC itself, indicating an auto-regulatory control scheme. Often, however, TFs that were predicted by Lisa to be MeC regulators were not represented by high MeC *z*-weights, and TFs with high MeC *z*-weight were not always found to have high Lisa scores (Fig. 5, Supplementary Data 2: MeC regulators). TFs predicted by Lisa but not represented by high MeC *z*-weight could be the result of TF activities being regulated through non-transcriptional mechanisms[48] or multiple TFs in a family having similar binding patterns but only a subset being the regulators[49]. TFs that have high MeC *z*-weights but low Lisa scores are most likely not well represented in the relevant cell types in available ChIP-seq data. In MeC-0, interferon response, *STAT1* is highly represented in the MeC *z*-weight and Lisa ranks *STAT1* as the top regulator, consistent with *STAT1* being known as the master regulator of the interferon response (Fig. 5a). Several lineage-defining TFs display the autoregulatory pattern, including *TCF4* in plasmacytoid dendritic cells (pDC) (Fig. 5b) and *XBP1* in B

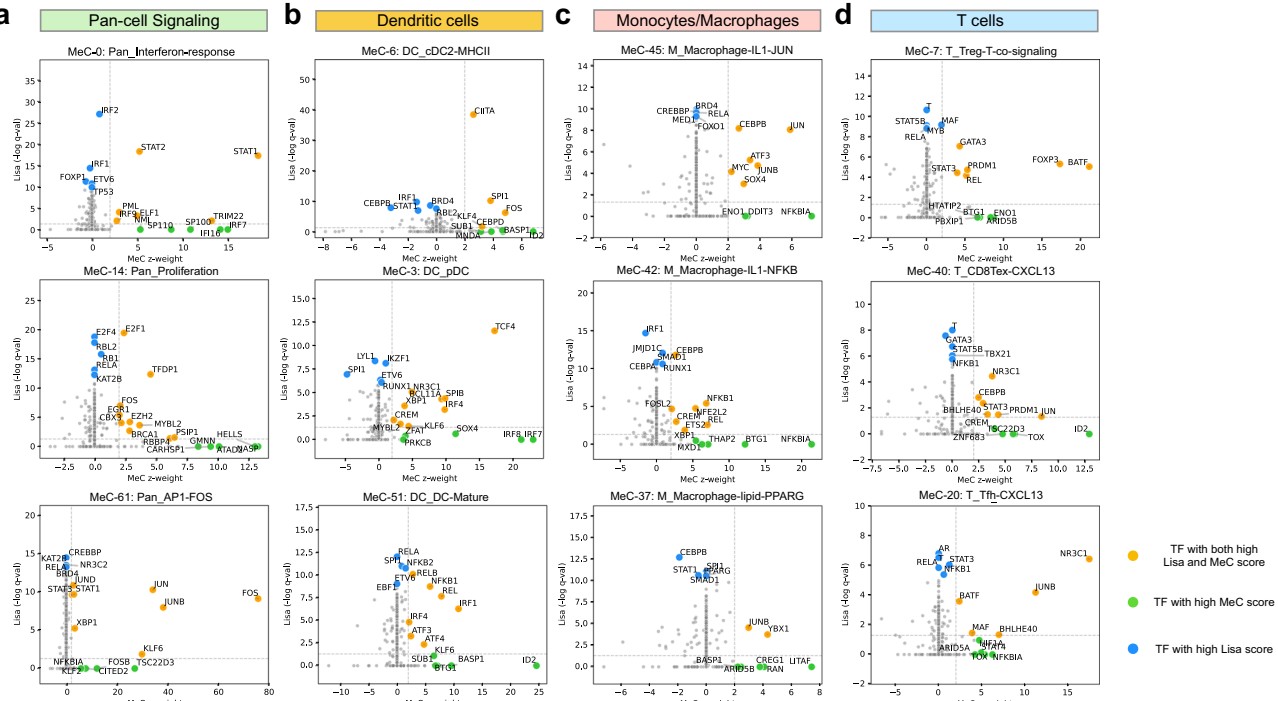

**Fig. 5 | MetaTiME prioritizes tumor immunity transcriptional regulators.** For selected MeCs, TFs are prioritized by their MeC expression representation and Lisa, ChIP-seq based, regulatory potentials. *X*-axis: gene *z*-weight of the TF for the current MeC. *Y*-axis: Lisa-based regulatory potential significance for top genes in the current MeC. Orange factors: MeC regulators prioritized based on both MeC gene weights and Lisa regulatory potential significance. Green factors: TFs highly weighted in MeCs and not in Lisa analysis. Blue factors: TFs with high Lisa regulatory potential and not highly weighted in MeCs. **a** TFs prioritized for three MeCs in the signaling category. **b** TFs prioritized for three MeCs in the dendritic cell category. **c** TFs prioritized for three MeCs representing different macrophage states. **d** TFs prioritized for three MeCs representing different T cell states. TF transcription factor, MeC meta-component. Source data are provided as a Source Data file.

plasma cells (Supplementary Data 2: MeC regulators). The macrophage related MeCs are regulated by myeloid lineage TFs like CEBPB, and TFs related to immune stimulus responses, including NFkB complex TFs. In MeC-37, lipid-rich macrophage, although PPARG ranks among the top Lisa-predicted regulators (Fig. 5c), *PPARG* expression is not highly represented as a MeC *z*-weight. In this MeC, the top co-expressed genes are indeed enriched in the PPARG signaling pathway (Fig. 4d); this result can be accounted for by PPARG being regulated through its ligands, which include a variety of lipophilic acids[48].

We found glucocorticoid receptor (GR) signaling to be implicated in the regulation of MeC-20, *CXCL13* + Tfh, with GR being most highly ranked TF in both MeC and Lisa scores (Fig. 5d). Top genes in MeC-20, *CXCL13* + Tfh, include several direct target genes of GR, including *SRGN* and *FKBP5* (Supplementary Fig. 13b). We investigated whether the *CXCL13* cytokine itself could be a direct target of GR in *CXCL13* secreting Tfh cells. Since GR ChIP-seq data is not available for the exact Tfh cell state, we collected GR ChIP-seq data from several other cell types. Direct binding of GR is observed at the *CXCL13* gene promoter and nearby the gene locus at putative enhancers, which are conserved across multiple cell lines (Supplementary Fig. 13a), including the B cell line Nalm6, the monocyte cell line THP1, and cancer cell lines. Moreover, in another CXCL13 secreting cell state, MeC-40, *CXCL13* + exhausted CD8 T cell, GR is also highly ranked in both Lisa and MeC scores (Fig. 5d). Thus, we hypothesize that GR is likely to be a transcriptional driver of the CXCL13-secreting cell states in exhausted CD8 T cells[50] as well as in CD4 T follicular helper T cells. Thus, the GR pathway could be a candidate target in tumor immunity modulation.

## Discussion

We developed the MetaTiME (Meta-components of the tumor immune Microenvironment) framework and performed a large-scale and pan-cancer integration of tumor single cell datasets using ICA to optimize information independence among components[51]. We identified 86 interpretable meta-components (MeCs) that describe common aspects of TME gene expression variation across multiple tumors.

The MetaTiME MeCs serve as comprehensive transcriptional signatures that depict a functional landscape of TME transcriptional programs and cell states. For monocytes and macrophages, the related MeCs showed heterogeneity and plasticity of tumor-associated macrophages (TAMs). We thus propose that TAMs, especially for solid tumors, should be classified based on the major states with different metabolic preferences instead of the canonical M1 and M2 classification[42]. Similar states that do not fit well into the M1 versus M2 classification scheme were also observed in previous studies analyzing myeloid cells in the TME, where single cells were clustered and labeled using differential markers[10,12,43]. Cheng et al.[10] defined several TAM types by clustering myeloid cells separately for each study and naming the TAMs with manually selected top marker, chosen based on consistency across cohorts. Similarly, Bi et al.[43] defined TAM types by harmonizing patients and naming the TAMs with top genes in each cluster for an advanced renal cell carcinoma (ccRCC) cohort. Cell type definitions in the previous studies were based on representative genes, which were chosen differently in the respective studies. We propose that the MetaTiME derived monocyte and macrophage MeCs could be used to define macrophage states and functional co-expressed gene modules more consistently for the TME. Reexamining the marker genes from the previous studies: *NLRP3* is highly ranked in MeC-42, *IL1B* + macrophage; in fact, the NLRP3 inflammasome mediates interleukin-1β production. *GPNMB* is weighted among the top 20 genes in both MeC-19, Macrophage-C1Q and MeC-26, Macrophage-SPP1; it encodes a membrane glycoprotein which is typically highly expressed in macrophages. *FOLR2* is ranked 29th in MeC-19, Macrophage-C1Q,

indicating this macrophage state also encodes a high folate-activated pathway. Finally, MeC-36, Macrophage-SPP1-C1Q, features an intermediate state with both SPP1 and C1QA, indicating the plasticity and mixed nature of pathways activated in TAMs that could not be defined using exclusive markers. Thus, the myeloid MeCs may provide a consistent definition of TAM states corresponding to different metabolic processes.

MetaTiME provides a toolkit for analyzing independent TME scRNA-seq datasets by mapping gene expression onto the MeC space. The outputs include signature continuums and the most highly enriched cell states. Recent useful single-cell dataset integration algorithms such as Harmony[22] and scArches[52] infer a joint low-dimensional representation among data. In these approaches the shared space is recomputed every time a new dataset is incorporated. The MetaTiME strategy builds upon previous approaches that transfer latent representations from large datasets, but provides a stable and interpretable representation specialized for the TME.

By leveraging ChIP-seq data, MetaTiME implicates critical transcriptional regulators in tumor immunity. In many cases, we found the joint consideration of MeC-specific co-expression patterns and TF binding enrichments shows the functions of TFs in defining cellular states and gene expression programs. MetaTiME captured multiple known TFs critical to tumor immunity and could serve as immune modulation targets; this includes TOX in MeC-40, CXCL13-secreting exhausted CD8 T cell, a recently discovered regulator of T cell exhaustion[53] (Supplementary Data 2). The MeCs further implicated the glucocorticoid receptor pathway in the regulation of several T cell states. Glucocorticoids are a class of steroid hormones essential to the modulation of multiple biological processes, including immune related ones[18], although the function of the GR pathway in different immune cell types is not fully understood. Since GR is broadly expressed in many cell types, and is regulated through ligand binding, differential analysis of GR expression is unlikely to fully capture GR regulation in single-cell data analysis. Though GR ChIP-seq is not available in the contexts of the relevant T cell states, GR ChIP-seq in other cell lines demonstrate robust binding nearby the top gene CXCL13. CXCL13 is crucial to T follicular helper cell communication with germinal center B cells, through interaction with its receptor CXCR5[54,55].

Overall, MetaTiME depicts the functional landscape of transcriptomic variation and cell states in the tumor microenvironment. It provides a computational framework to facilitate the elucidation of the identity and function of cells in the TME in future studies and will facilitate the identification of potential new therapeutic targets for immune modulation.

## Methods

### Tumor single-cell RNA-seq data collection and processing
For an extensive collection of single cells from tumor microenvironment, we utilized the public tumor scRNA-seq collection, TISCH[28]. The TISCH collection uniformly processed each dataset with MAESTRO[17] and isolated non-malignant environmental cells from malignant cells. Overall, we collected 2,157,387 cells from 76 studies ranging 27 cancer types. The MAESTRO annotation was labeled using CIBERSORT gene panels[18] followed by curation, enabling selection of 1,719,813 tumor micro-environment cells, including 1,617,110 immune cells and 102,703 stromal cells, were retained for integrative analysis in this study. For studies with data measured from multiple cancer types, cells different cancer types were split into independent datasets, resulting in 93 datasets; it includes 3 PBMC datasets from healthy donors from 10X Genomics as baseline and 7 datasets with ICB treatment.

For an unsupervised component analysis, each dataset was reanalyzed. For datasets with raw count matrices available, gene expression was normalized towards per-cell read depth 10,000 followed by log transformation. For datasets with only TPM or FPKM values available, including Smart-seq data or studies with only normalized matrix available, gene expression underwent log transformation. Cells were filtered based on minimum library size 1000, gene number 500, and maximum mitochondrial read proportion 5%.

### Decomposing individual studies and denoising low-dimensional components
We then decomposed the expression matrix of each scRNA-seq dataset using fastICA[51] into an independent component (IC) vector matrix and a projection weight matrix. We tested different values for the number of components ($k$) and chose $k$ to be 100 uniformly for each dataset, given it could cover more variations than the number of cell types in the TME, which is around twenty. We applied two denoising approaches to handle sparsity and potentially noisy ICs. First, we performed a $z$-score transformation of the gene loadings in the component, scaling all gene loading values by the standard deviation of each IC. The gene loadings indicate the degree of contribution to the component as a metagene from each gene, and we observed that most genes contribute neutrally to the metagene. Thus, genes with significant contributions are selected using the two-standard deviation threshold from either the positive or the negative side. Second, we aligned the positive skewness of components (using scipy.stats.skewtest[56]) since the sign of an independent component is randomly assigned in fastICA optimization. We observed that asymmetrically extreme gene loading values highlight genes representative of the component's function; thus, we computed each component's skewness statistics and flipped the sign of component loadings if the skewness is negative. We excluded genes with a low contribution (gene weight not passing two standard deviations) to any reproducible components and kept 6623 genes with potentials in driving the reproducible components. The post-decomposition steps ensured the attitude and sign of gene weights are comparable across cohorts, depicting degree of contribution from each gene in the genome-wide background.

### Meta-component calling and functional annotation
We then aim to discover reproducible patterns from all components from each dataset. We evaluated similarity between pairs of components using cosine distance and retained a set of 1043 candidate reproducible components from 69 datasets, each with a minimum Pearson correlation coefficient 0.3 with at least one different IC. We then clustered ICs using Louvain clustering, a graph-based community detection algorithm where the resolution parameter controls segmentation granularity. Clusters with at least five ICs were retained as reproducible IC clusters. The number of clusters is determined by optimizing both Silhouette's score for optimal within-cluster similarity compared to inter-cluster similarity and number of reproducible clusters. The final resolution parameter was chosen to be 1.25 resulting in 86 clusters for meta-component (MeC) calling. The consensus gene $z$-weights in each MeC were then calculated by averaging ICs in each cluster. Genes of outlier $z$-weights passing two standard deviations were highlighted as significant, and the ones with positive largest $z$-weights were considered representative of the MeC.

MetaTiME MeCs were assigned curated annotation by matching top $z$-weighted genes to functional biological information including cell type markers, pathway databases from GSEA and Enrichr through GSEApy, cell types expressing top MeC genes, and high-rank transcription factors. In detail, GSEA enrichment analyses utilized the top 100 highest $z$-weighted genes and TF database was obtained from AnimalTFDB[49]. The pathway database from Enrichr utilized WikiPathways, BioPlanet, MSigDB_Hallmarks, GO_Biological_pathway, GO_Molecular_function, GO_Biological_process. KEGG, and Reactome. GSEA was also performed using the $z$-weight rank of all genes[57]. The 86

MeCs were first ordered by MeC cluster size, then organized into seven functional categories with six lineage-related categories and one signaling category.

## Simulating multi-cohort single-cell RNA data with expression programs

To benchmark dimensional reduction methods, we built upon previous effort from Kotliar et al.[58] to use the scsim package to simulate multiple count matrices with built-in transcriptional programs. In principle, the built-in gene expression programs (GEP) were sampled as random scaling factors on a subset of genes mimicking overexpression or suppression of a pathway. For testing whether a higher number of cohorts facilitate GEP recovery, we simulated 20 single-cell datasets and tested usage of 5 cohorts, 10 cohorts, and 20 cohorts. Each dataset was embedded with a subset of 14 pre-defined GEPs, since the real tumor scRNA data may not cover every possible cell type or gene program in every dataset. The 14 GEPs contain 13 cell type-specific programs with distinct cell type-specific genes, and one signaling gene expression program that is randomly active in multiple cell types. Two low-dimensional reduction method are benchmarked using simulated scRNA data: independent component analysis (ICA) and non-negative matrix factorization (NMF). Decomposition was performed on each single cohort separately, and meta-component calling was done as similar in MetaTiME: components are filtered, clustered into meta-components, followed by averaging gene $z$-weights per cluster as predicted gene expression programs (GEP). The predicted GEPs were compared with pre-defined True GEPs using Pearson correlation. Overall, both ICA and NMF can recover GEPs, while the ICA-based GEPs are more mutually independent and performs slightly better. Since the GEP recovered in the 20 cohorts case matched true GEP better than 5 cohorts and 10 cohorts, the increased number of cohorts also improves GEP recovery. Thus, we chose ICA for component integration and use all available datasets for GEP discovery for tumor microenvironmental cells.

## MeC calling robustness with leave-one-out testing

MetaTiME's robustness was examined by re-calling MeCs after leaving one dataset out and calculate the Pearson correlation between the left out dataset MeCs and the full-set MeCs. The leave-one-out analysis was first performed by removing the SCC cells from the Yost et al. dataset, which was shown as an example of MetaTiME annotation. Each dataset in the MeC pool was left out to call new MeCs using the same parameters, resolution = 1.25 and min_cluster_size = 5. We used two metrics to check the similarity of each leave-one-out experiments compared to the full set. The first metric is the mean maximum correlation between leave-one-out and the full set. This is calculated by taking the maximum correlation, row-wise and column-wise in the correlation heatmap, and then taking the average. The second metric is the number of MeCs.

## Comparing MeCs, signatures by post-embedding integration, to signatures by cluster-wise differential expression

We compared MetaTiME MeCs to the standard approach that extracts signatures after integrating cells across datasets. The standard approach to obtain cluster-wise signatures is to perform dataset harmonization, cell clustering, and cluster-wise differential expression analysis. When performing dataset integration, we first used CCA to include as many datasets as possible given our maximum memory available in our computing resource (150GB). The maximum number of datasets integrated by CCA is 10 when using datasets ranked by the number of TME cells. We then used scanpy to achieve the successful integration of 21 datasets. The integrated cells are harmonized with Harmony[22] and clustered, followed by differential expression analysis using the Wilcoxon test to extract cluster-wise signatures. The signatures are in the format of gene vector of log fold change, and named as "Cluster DE signatures". To control the confounding effect of

signature numbers, the resolution of clustering is chosen to reach 90 clusters, a similar number with the number of full-set MeCs which is 86. Then, we compared MetaTiME MeCs to the Cluster DE signature projected onto a test dataset. We used same BCC cells as the test cells that are never seen in the training step from either method. To reflect how variable the projected scores are across cells, we grouped cells from the test BCC data into 25 clusters with Leiden clustering. The heatmap of the cluster-wise signature mapping is plotted to observe how specific each type of score is across cell groups. We further used cross-cluster entropy to quantify the information content of scores relative to clusters in the test dataset.

## The MetaTiME annotator for analyzing new tumor scRNA-seq data

MetaTiME provides an analytical toolkit for annotating cell states and signature activities for tumor scRNA-seq data (https://github.com/yi-zhang/MetaTiME). The scRNA-seq data is first processed following standard procedures, which includes cell depth normalization, *log*-transformation, batch effect removal using Harmony[22], neighboring graph construction, graph clustering, and UMAP embedding for visualization. Specifically, the clustering step uses an over-clustering strategy, which sets a high-resolution parameter (default 8) that generates a larger number of clusters and help reveal fine structures among the cells. Then, the MetaTiME annotator tool takes as input a single log-transformed expression matrix for TME cells from the dataset. The outputs include both per-cell MetaTiME MeC signature scores and per-cluster enriched MeC state. For the per-cell score, MetaTiME projects each cell onto the MeC space by calculating the dot product between the expression vector and the $z$-weight vector of each MeC, using genes passing the significant $z$-weight criterion ($z$-weight $\geq 2$). The projection matrix is then scaled across all cells to ensure normally distributed scores within each MeC, outputting the cell-by-MeC score matrix. Meanwhile, the UMAP view of the projection score shows the signature gradient across the cells positioned by similarity. Lastly, the cluster-wise MeC enrichment results are also generated. The per-cluster MeC enrichment score is calculated by averaging profile of cells along each MeC; MeCs with mean score passing the significant cutoff (2 in the $z$-weight scale) are called as the set of enriched MeCs. Each cluster may enrich multiple number of MeCs, and the top enriched MeC with highest score is used in UMAP visualization.

## Differential MeC signature analysis

For tumor scRNA-seq data with different conditions, a differential signature analysis can be carried out following MetaTiME annotation, which provides enriched MeCs for each cluster and names each cluster with the top enriched MeC. Thus, for each cell cluster, the MeC signature strength can be compared across conditions, for all enriched MeC in the current cluster. In details, a simple $t$-test or Wilcoxon rank-sum test is adopted to compare MeC scores of cells in one condition with another. The effect size of MeC scores were calculated by the difference between cell means from the two conditions in comparison. To plot the cluster-specific differential signature plot, the signatures are marked using "EnrichedMeC@ClusterName", where the "ClusterName" is the top first enriched MeC used as cell state as current cluster. When "EnrichedMeC" is the same as the "ClusterName", only "ClusterName" is marked on the Significance-Effect size plot.

## Incorporation of epigenetic data using Lisa

Our group previously developed Lisa that predicts the influence of TFs on a set of genes. Lisa models public chromatin accessibility and TF binding profiles to score TFs in gene regulation from an epigenetic perspective. We developed Lisa2 that improves on running speed and pipeline integration, which is applied on each MeC to score TFs in regulation potential on top 100 high $z$-weighted genes. The impact

scores of TFs are thus from two sources: MeC z-weights for expression representation, and Lisa scores for binding potential, as in Supplementary Data 3. The TFs are grouped into three classes, TFs highly ranked based on both MeC gene weights and Lisa significance, TFs representative only in MeC, and TFs based on binding information only. In Supplementary Data 2, we marked TFs from different classes in different columns. Significant TFs based on both MeC and Lisa (MeC z-weight ≥2), Lisa score -log (p-value) ≥2 are marked in the column TF_MeCLisa_top_1; furthermore, the TFs ranking among top 40 (aggregated rank of MeC and Lisa) compared to all genes are further marked in the column TF_MeCLisa_top. TFs ranking among top 10 only in MeC z-weight are in the column TF_MeC_top, and TFs ranking among top 10 only in Lisa score are in the column TF_Lisa_top.

### Reporting summary

Further information on research design is available in the Nature Portfolio Reporting Summary linked to this article.

### Data availability

The processed datasets used in this study are available in Zenodo under accession code 7410180 The pretrained meta-components for tumor microenvironment in the Github repository: https://github.com/yi-zhang/MetaTiME. The single-cell RNA-seq data used in this study are available in the TISCH database (http://tisch1.comp-genomics.org). The list of public datasets used in this study is available in Supplementary Data 1 and also from GEO under accession code GSE154763 and Single Cell Portal under accession code SCP1288. The gene list analytical data used in this study are available in AnimalTFDB v3.0 (http://bioinfo.life.hust.edu.cn/AnimalTFDB/#!/), TCGA (https://portal.gdc.cancer.gov), and Azimuth (https://azimuth.hubmapconsortium.org). The ChIP-seq data used in this study are available from GEO (https://www.ncbi.nlm.nih.gov/geo/) under accession IDs GSM604651, GSM1637309, GSM1607526, GSM2661793, GSM2735378, GSM2871705, GSM1637306, GSM1637307, and from ENCODE (https://www.encodeproject.org/) under accession ID ENCSR919OXR. Source data are provided with this paper.

### Code availability

MetaTiME is available at https://github.com/yi-zhang/MetaTiME, and also available at Zhang Y et al., "MetaTiME Integrates Single-cell Gene Expression to Characterize the Meta-components of the Tumor Immune Microenvironment", MetaTiME, https://doi.org/10.5281/zenodo.7734062, 2023.

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

## Acknowledgements

The results appearing here are supported by grants NIH grants P01CA163222(CM), U24CA237617(CM,MB), and R01CA234018(MB). We acknowledge all authors of the original tumor single-cell studies that generated publicly available data. The results appearing here are in part based upon the data generated by the TCGA Research Network (http://cancergenome.nih.gov/).

## Author contributions

C.M., M.B., S.L., Y.Z. designed the study and were major contributor in editing the manuscript. C.M., M.B., S.L. supervised the study. Y.Z. analyzed and interpreted the data and was a major contributor in writing the manuscript. Y.J., G.X., C. Wan, Z.Z., A.L., W.Z., J.F., J.K. contributed to the data analysis. C. Wang, Z.Z., Y.Z., Y.J., S.G., B.Z. contributed to the data collection and analysis. All authors read and approved the final manuscript.

## Competing interests

MB is a consultant to and receives sponsored research support from Novartis. MB serves on the SAB of H3 Biomedicine, Kronos Bio, FibroGen, and GV20 Therapeutics. X.S.L conducted the work while being on the faculty at DFCI, and is currently a board member and CEO of GV20 Therapeutics. The remaining authors declare no competing interests.
