## [Peer Review File · Nature Communications]

MetaTiME Integrates Single-cell Gene Expression to Characterize the Meta-components of the Tumor Immune MicroenvironmentREVIEWER COMMENTS

Reviewer #1 (Tumor immunology, scRNAseq) (Remarks to the Author):

In Zhang et al, the authors describe MetaTiME, a framework for assessing the TME where large datasets across studies were integrated to learn components of the TME. The integration and establishment of the metacomponents (MeCs) that represent signatures in the aggregate data are interesting, and the ability for a potential user to map their own data onto the MeCs could be very useful. The study itself could use some additional analysis and clarifications and generally would benefit from additional head-to-head comparisons with other methods currently used to accomplish similar tasks (or portions of tasks within the MetaTiME framework).

Comments:

The authors perform their integration, yet there is no comparison to other dataset integration tools. A comparison to CCA for cluster identification would be useful. I recognize that the goal here is slightly different in that signatures are being pulled out, but ultimately it is quite similar. A more detailed portrayal of which ICs are present and in which studies would be useful as well.

More detail should be provided in the main text regarding the IC filtering (lines 110-111), also which graph clustering algorithm? (put in main text that it is Louvain)

Again a comparison of the annotated cell states using MetaTiME vs something like CCA->clusters is important. Why is MetaTiME better than taking all these datasets, integrating, calling clusters, and finding top marker genes for each cluster? I believe the authors can make a compelling case, but that case needs to be made by performing direct comparisons versus discussing theoretical differences in the framework alone.

The comparison using Seurat->CIBERSORT, etc... for cell type annotation is useful for annotation of de novo datasets. It would also be helpful to include some metrics as opposed to qualitative observations (eg "higher-resolution"). However, was the Yost et al dataset used in the integration and training of the MeCs? If so – this needs to be redone with that dataset removed. Ideally a full leave-one-out analysis would be the best option in order to show performance on multiple datasets when those are not used in the training set. (If Yost et al was indeed excluded from the training dataset, it should be directly stated in the text)

The above comment also goes for the ICB analysis. While there is value in assessing all data together for better understanding ICB; using a leave-one-out to assess performance on de novo datasets would be valuable proof-of-principle of MetaTiME as a tool for annotation and assessment of new datasets.

In the myeloid metabolic section – this is also in the void of any comparison to other methodologies (e.g. integration and clustering using standard methods). If the focus is the tool itself, then the tool needs to be directly compared to other methods and show benefit. The biology uncovered is interesting and there is novelty in the integration of such a large set of studies, but there is currently no direct evidence that it could not have been done using standard approaches.

Minor:

Line 57 – typo – “approach is TO use”, sentence is a bit long. Maybe just “Another approach is to map a dataset containing unannotated cell states onto an annotated reference” or something like that.

Lots of text in the figures is too small

Reviewer #2 (Machine learning, computational analysis) (Remarks to the Author):

In their manuscript "MetaTiME: Meta-components of the Tumor Immune Microenvironment", Zhang et al. describe a novel tool for studying the tumor microenvironment in single-cell RNA-seq data. Studying the TME is of great importance to understand the response of the immune system and to inform about immunotherapy and treatment response. Single-cell RNA-seq data offer a wealth of information about the TME but a bottleneck in the analysis is the annotation of individual cell types and cell type clusters. While a plethora of methods exists for cell type annotation these can only offer crude annotations and rely on pre-defined marker genes, neglecting most of the cell-type-specific information of the transcriptome. With every new data set, the annotation task begins anew which is why efforts of integration and joint annotation with tools such as Harmony or scVI offer a joint embedding where cell clusters can be uniformly annotated. An limitation of this approach is that the embedding is typically started from scratch whenever a new dataset is added. The aim of MetaTiME is thus to obtain a stable embedding in which new datasets can be easily projected, thus simplifying the annotation of cell types in the TME. Second, MetaTiME strives for offering insights into fine-grained cell types going beyond classical coarse annotations, e.g. going beyond M1 and M2 subtypes for macrophages. A third aim is to offer a functional readout, giving insights into the activity of these fine-grained cell types in different data sets and tumor types. MetaTiME achieves these goals in a simple and elegant fashion. First, a large number (1.7 mio) of single-cells from 79 datasets spanning different cancer types has been collected. Malignant cells have been removed to focus on cells in the TME. The remaining cells were integrated using independent component analysis (ICA) which the authors found to outperform non-negative matrix factorization. ICA has the advantage that it delivers feature weights for each gene contributing to a component which allows measuring similarity of the components. IC vectors of individual data sets are z-scored and skewness aligned to become comparable. Subsequently, these are clustered to remove redundancy and to obtain a cross-data-set representation. This was achieved using the Louvain method where scores were aggregated with the mean gene scores to obtain a final set. With the z-scores the resulting components are easily interpretable by looking at top-scoring genes or by performing GSEA. New data sets can be projected (after clustering) into this space by computing the dot product of the gene expression values and the z-score weights of the components which is an elegant way of obtaining an annotation. The authors show in simulations and across multiple data sets that MetaTiME offers good interpretability (also by adding insights into regulating transcription factors through the LISA method) and a good resolution on cell types (and subtypes) relevant for understanding the TME. The manuscript is well written and easy to follow. The method is straightforward but innovative and a great addition for the community. I have only few comments:

#Major:

- MetaTiME offers differential signature analysis (difference of MeC component activity between conditions). However, it is not clear if differential abundance analysis is also supported (this would be going back to counting individual cells after annotation). This may be useful since the (relative) abundance of cell types is an important factor.
- Most steps needed to reproduce the method are clearly explained but the GSEA analysis lacks details. It is not clear which exact method was used (incl. details such as the background used). I also wonder why only the first 100 genes were considered as GSEA naturally considers the ranks of all genes (in contrast to gene set overrepresentation analysis / hypergeometric test which is sometimes also called GSEA and should not be confused).
- To be honest I did not understand the part of the skewness of the scores. Skewness in one direction apparently informs about the directionality of the underlying function and the associated genes. At least I (probably others) would benefit from a supplementary figure where this is shown for a few selected examples.
- For future research it would be an interesting perspective to check if the components inferred in MetaTiME could be used as a source of signatures for the deconvolution of bulk RNA-seq data into more fine-grained cell types.

Reviewer #3 (scRNAseq, systems biology) (Remarks to the Author):

Zhang et al. develop MetaTiME, a pan-cancer reference set of cellular identities and states for tumor microenvironment which can be used to annotate new datasets with continuous scores.

This is an area of active research and the general concept brought forward by the authors, while not entirely novel conceptually is enticing.

The manuscript is clearly written and the authors provide well written, documented and working software implementation.

However, I have a few concerns about the interpretation of the latent variables discovered by the authors across TMEs, and on how generalizable the method is for broad use by new users.

Below I provide a series of inquiries and recommendations to the authors that hopefully will help improve the manuscript:

On the methodological approach:

- The authors show that MetaTiME provides more resolved cell labels for an existing dataset which I assume is also used in the ICA training of the meta-components. One issue with that is that given enough time or effort, the authors of the original dataset could likely have generated a similarly granular annotation as well. In order to demonstrate the usefulness of MetaTiME in annotation of a broad set of tumor data, I propose the authors work with data that has not been seen for training. This could be accomplished by cross validation in a leave one cancer out strategy, in which all but one cancer type is used in training and then the left out cancer is annotated for the first time. The scores for TME cells in that held out cancer can then be compared with the scores in the fully trained model for evaluation, and this done for every cancer type. A second approach would be that the authors either generate data from a new cancer type or use a yet unused dataset from a novel cancer type for this purpose. I believe this would be important to demonstrate the usefulness of MetaTiME in annotating datasets in a much more real situation for new users.

- The authors claim batch effects are negligible by requiring MetaTiME meta-components to have samples from more than one batch. This is a positive step, albeit a relatively weak one as it does not guarantee that the sample structure in or across datasets is still not confounded by non-obvious sample groupings or even incorrect experimental designs in the original datasets. Could the authors explicitly compute the enrichment of different batches for each dataset in each meta-component, or complementarily do differential testing of meta-component scores between batches? This would reinforce that the discovered components reflect biology and not technical artifacts.

- The authors provide no systematic validation of their interpretations of the meta-components, but however chose to give ample specific examples of genes enriched in signatures as justification for naming meta-components. While there is value in mentioning specific examples, the lack of a very systematic comparison of the proposed annotation for the metaclusters prevents the reader from assessing how good (or not so good) in general the provided interpretation is. It would therefore be great to assess how well the author's interpretations of the components' weight matches literature. One way could be for example to take the top N terms from each component and doing enrichment analysis with Enrichr for gene set libraries related with cell types, pathways, transcriptional regulators, and providing the relative enrichment in all of these as an additional tensor of information to be explored by the users of MetaTiME, and presented as validation of the manual annotation by the authors by observing how often each chosen term is recovered in the top enriched terms. In addition, it would be better if the nomenclature of the components were more systematic for example by always having first a broad cell type name, then a specific cellular state, as well as standardizing the capitalization and abbreviation (for example by not using abbreviations) of the meta-components.

On data availability and software implementation:

- Thank you for making the software freely available to everyone.

- I commend the authors for the clearly documented software and availability of Jupyter notebooks exemplifying the usage of the software.

- Please make sure to make available the preprocessed datasets used in creating the TME reference as well in a publicly accessible database (e.g. Zenodo) to enhance reproducibility of the manuscript and reuse of resources across the community.

On the figures:

- Figure 3d is not mentioned.

RESPONSE TO REVIEWER COMMENTS

Point by point responses to the reviewers' comments on “**MetaTiME: Meta-components of the Tumor Immune Microenvironment**” are included below (reviewer comments in blue, our responses in black).

Reviewer #1 (Tumor immunology, scRNAseq) (Remarks to the Author):

In Zhang et al, the authors describe MetaTiME, a framework for assessing the TME where large datasets across studies were integrated to learn components of the TME. The integration and establishment of the metacomponents (MeCs) that represent signatures in the aggregate data are interesting, and the ability for a potential user to map their own data onto the MeCs could be very useful. The study itself could use some additional analysis and clarifications and generally would benefit from additional head-to-head comparisons with other methods currently used to accomplish similar tasks (or portions of tasks within the MetaTiME framework).

We thank the reviewer for carefully reading our manuscript and pointing our areas where it might be strengthened. We have made the requested comparisons (details provided below).

Comments:

1. The authors perform their integration, yet there is no comparison to other dataset integration tools. A comparison to CCA for cluster identification would be useful. I recognize that the goal here is slightly different in that signatures are being pulled out, but ultimately it is quite similar. A more detailed portrayal of which ICs are present and in which studies would be useful as well.

We thank the reviewer for the suggestion to compare MetaTiME to CCA-based signature extraction. We first attempted to use CCA to include as many datasets as possible using the maximum memory available in our computing resource. On a server with 150GB memory, CCA would run out of memory on only 10 datasets, crashing after 3 days. We thus turned to scanpy, using harmony to integrate as many datasets as possible, where the maximum number of datasets integrated can reach 21 successfully. Using the 21 datasets with highest number of TME cells, we generated signatures to compare to MetaTiME.

We have included a new supplementary figure for this test (Fig.S6). As shown in Fig.S6a, the batch effects from different datasets can be corrected by Harmony integration. We then performed Leiden clustering on the integrated map. For fairness of comparison, to reach a similar number of signatures as MetaTiME MeCs, the clustering resolution was tuned to achieve 90 clusters (resulting in a parameter value of 4). We then performed differential expression using the default Wilcoxon test and obtained a list of “Cluster DE signatures” (Fig.S6b). Each Cluster DE signature is a full vector of per-gene log fold change comparing the corresponding cluster to all the rest cells in other clusters, for all 10702 genes.

We compared MetaTiME MeCs to the Cluster DE signatures projected on the test data using the same basal cell carcinoma (BCC) dataset as was used in the original submitted manuscript. This dataset was never seen in the training step from either method. As the gold standard of cell states is not available, we assume that a good set of cell state signatures learned from training datasets is more cluster-specific rather than evenly distributed across all cells when projecting

on a testing dataset. The specificity can be quantified by calculating the entropy of mean signature values across cell clusters in test dataset. This analysis assesses the unevenness of the signature score distribution among clusters in the test dataset.

We grouped cells from the test BCC data into 25 clusters with regular Leiden clustering using default parameters (Fig.S6b, middle). We reasoned that different cell types and reproducible cell states would underlie the gene expression programs defining the Leiden clusters, and that a good representation of cell types/states would show a high level of specificity to different clusters. To reflect the variability of projected scores across cell clusters we plotted a heatmap of cluster-wise signatures. The MetaTiME MeC scores are more specific to cell clusters compared to the Cluster DE signature scores (Fig.S6c, top). We further used entropy to quantify the specificity of scores to clusters in the test dataset, plotting the entropy of each score (Fig.S6c, bottom). The MetaTiME MeCs generally have lower entropy compared to the Cluster DE scores, meaning that MetaTiME scores are less uniformly distributed across test data clusters compared to Cluster DE signatures.

We have included a description of this analysis in the main text (lines 112-128). We have also added a new supplementary figure Fig.S6, and reordered the indices of other supplementary figures. We have added a new section in the Methods section: “Comparing MeC signatures by post-embedding integration, to signatures by cluster-wise differential expression.”

To portray which independent components are present in which studies: we have edited the main text line 119 to refer to FigS4 with a coloring of cohort sources under the heatmap for all ICs in the heatmap columns. We have included the file with detailed independent component names in each MeC in our github code depository as a part of the pretrained MeCs.

2. More detail should be provided in the main text regarding the IC filtering (lines 110-111), also which graph clustering algorithm? (put in main text that it is Louvain)

Following the reviewer’s suggestion; we have edited the text on current lines 115-117 to reflect the method details.

“Next, MetaTiME filters ICs to retain ones that are reproducible across multiple cohorts (the minimum Pearson correlation with any other IC ≥ 0.3). These are passed to a graph clustering algorithm, Louvain clustering, to merge IC groups into MeCs (Methods).” This detail is also included in Methods section.

3. Again a comparison of the annotated cell states using MetaTiME vs something like CCA->clusters is important. Why is MetaTiME better than taking all these datasets, integrating, calling clusters, and finding top marker genes for each cluster? I believe the authors can make a compelling case, but that case needs to be made by performing direct comparisons versus discussing theoretical differences in the framework alone.

The new analysis comparing MetaTiME signatures with an alternative standard approach, as described in the response to point 1 address this point too. In this test for found that the cluster-wise signatures are not compatible with large number of datasets, which will become increasingly problematic as more data accumulates. Furthermore, the cluster-wise signatures display lower specificity in the test data. To improve the manuscript reflecting the comparison, we have also edited the main text in lines 122-128.

4. The comparison using Seurat->CIBERSORT, etc... for cell type annotation is useful for annotation of de novo datasets. It would also be helpful to include some metrics as opposed to qualitative observations (eg "higher-resolution").

As suggested, besides the figure comparison between MetaTiME annotation and CIBERSORT annotation, we have added the description of the number of annotated cell types/states in line 201-202. "In addition, compared to the Seurat's¹⁴ automated CIBERSORT marker-based annotations (14 cell types, Fig.S8a), MetaTiME provides higher resolution (38 cell states, Fig.3b)."

5. However, was the Yost et al dataset used in the integration and training of the MeCs? If so – this needs to be redone with that dataset removed. Ideally a full leave-one-out analysis would be the best option in order to show performance on multiple datasets when those are not used in the training set. (If Yost et al was indeed excluded from the training dataset, it should be directly stated in the text)

We thank the reviewer for pointing out this important point. Indeed, we did not include the cells from Yost et al. as shown in Figure 3. The Yost et al. data contain tumors of two cancer types, Squamous Cell Carcinoma (SCC) and Basal Cell Carcinoma (BCC). In the current MetaTiME, we used the cells from SCC tumors and excluded cells from the BCC tumors in the training. Thus, the "test cells" in Figure 3 are from BCC cells that were never seen by MetaTiME in the training steps. We have edited the text in line 196 to clarify the training-test separation. "We demonstrate the application of MetaTiME on basal cell carcinoma (BCC) single-cells from Yost et al. These test cells were excluded from the MetaTiME training stage."

6. The above comment also goes for the ICB analysis. While there is value in assessing all data together for better understanding ICB; using a leave-one-out to assess performance on de novo datasets would be valuable proof-of-principle of MetaTiME as a tool for annotation and assessment of new datasets.

We thank the reviewer for suggesting the leave-one-out experiments. We checked whether further leaving out SCC from the full *Yost et al.* ICB dataset would have an impact on the meta-components, we removed the SCC and performed the leave-one-out MeC calling step. We added a supplementary figure for the leave-one-out test. As shown in Fig. S5, MeCs called from the full set and the MeCs called from leave-SCC-out have high correlation and high specificity. There is one parameter, the minimum number of ICs in one MeC, that can affect the MeCs after leaving one dataset out. When keeping the number of minimum ICs in one MeC to be 5, there were two MeCs missing in the leave-SCC-out MeCs (Fig S5a), because of two MeCs that were contributed from the SCC datasets. If we reduce the minimum number of datasets supporting a MeC to be 4, the two missing MeCs are rescued (Fig S5b). In summary, the leave-one-out MeCs are highly correlated to the full-dataset MeCs based on the high correlation in both heatmaps (Fig S5a,b). This is an advantage brought about by the integration of a large number of datasets, which will not be achievable by the standard way of integrating cells due to limitations in computing resources and time availability.

As suggested, we further performed leave-one-out tests for every dataset. We still used the same parameter of cluster calling resolution and the minimum IC number in one cluster to be 5. We used two metrics to compare each leave-one-out experiment with the full set. The first metric is the mean maximum correlation between leave-one-out and the full set. This is calculated by taking the maximum correlation, row-wise and column-wise in the correlation heatmap, and then take averaging. This metric reflects the strength of the correlation along the

diagonal in the MeC correlation heatmaps. The second metric is the number of MeCs. As shown in FigS5c, the mean max correlation is high in each leave-one-out test (mean correlation=0.988). Similarly, the number of MeCs from each leave-one-out test is similar to the full set: mean MeC number across leave-one-out test is 84.64, close to the full set MeC number 86.

In summary, we made the following improvement of the manuscript: we added one supplementary figure Fig S5, edited the manuscript in line 115-120, and added one section in Method “MeC calling robustness with leave-dataset-out testing.”

7. In the myeloid metabolic section – this is also in the void of any comparison to other methodologies (e.g. integration and clustering using standard methods). If the focus is the tool itself, then the tool needs to be directly compared to other methods and show benefit. The biology uncovered is interesting and there is novelty in the integration of such a large set of studies, but there is currently no direct evidence that it could not have been done using standard approaches.

Similar to our response to comment 1, we added the comparison analysis to compare to a standard approach. In standard approaches, the number of clusters is affected by clustering resolution, the clustering is affected by the embedding space specific to the way of patient harmonization, clusters are named by picking one or two arbitrary markers, and definition of cell states are different across studies with minimum generalization. By integrating large set of studies and using unbiased combination of genes, we provide one solution of consistent definition of myeloid cell states.

Minor:

Line 57 – typo – “approach is TO use”, sentence is a bit long. Maybe just “Another approach is to map a dataset containing unannotated cell states onto an annotated reference” or something like that.

We have edited the text correspondingly in line 57-59.

Lots of text in the figures is too small

We have increased font sizes in the following panels considering the MeC name length: Fig. 2a,b, Fig. 3a, Fig. 4, Fig. 5.

Reviewer #2 (Machine learning, computational analysis) (Remarks to the Author):

In their manuscript "MetaTiME: Meta-components of the Tumor Immune Microenvironment", Zhang et al. describe a novel tool for studying the tumor microenvironment in single-cell RNA-seq data. Studying the TME is of great importance to understand the response of the immune system and to inform about immunotherapy and treatment response. Single-cell RNA-seq data offer a wealth of information about the TME but a bottleneck in the analysis is the annotation of individual cell types and cell type clusters. While a plethora of methods exists for cell type annotation these can only offer crude annotations and rely on pre-defined marker genes, neglecting most of the cell-type-specific information of the transcriptome. With every new data set, the annotation task begins anew which is why efforts of integration and joint annotation with tools such as Harmony or scVI offer a joint embedding where cell clusters can be uniformly

annotated. An limitation of this approach is that the embedding is typically started from scratch whenever a new dataset is added. The aim of MetaTiME is thus to obtain a stable embedding in which new datasets can be easily projected, thus simplifying the annotation of cell types in the TME. Second, MetaTiME strives for offering insights into fine-grained cell types going beyond classical coarse annotations, e.g. going beyond M1 and M2 subtypes for macrophages. A third aim is to offer a functional readout, giving insights into the activity of these fine-grained cell types in different data sets and tumor types. MetaTiME achieves these goals in a simple and elegant fashion. First, a large number (1.7 mio) of single-cells from 79 datasets spanning different cancer types has been collected. Malignant cells have been removed to focus on cells in the TME. The remaining cells were integrated using independent component analysis (ICA) which the authors found to outperform non-negative matrix factorization. ICA has the advantage that it delivers feature weights for each gene contributing to a component which allows measuring similarity of the components. IC vectors of individual data sets are z-scored and skewness aligned to become comparable. Subsequently, these are clustered to remove redundancy and to obtain a cross-data-set representation. This was achieved using the Louvain method where scores were aggregated with the mean gene scores to obtain a final set. With the z-scores the resulting components are easily interpretable by looking at top-scoring genes or by performing GSEA. New data sets can be projected (after clustering) into this space by computing the dot product of the gene expression values and the z-score weights of the components which is an elegant way of obtaining an annotation. The authors show in simulations and across multiple data sets that MetaTiME offers good interpretability (also by adding insights into regulating transcription factors through the LISA method) and a good resolution on cell types (and subtypes) relevant for understanding the TME. The manuscript is well written and easy to follow. The method is straightforward but innovative and a great addition for the community. I have only few comments:

We thank the reviewer for carefully reading our manuscript and a constructive critique.

#Major:

1. MetaTiME offers differential signature analysis (difference of MeC component activity between conditions). However, it is not clear if differential abundance analysis is also supported (this would be going back to counting individual cells after annotation). This may be useful since the (relative) abundance of cell types is an important factor.

We agree that differential abundance analysis of cell types is a useful analysis that is widely used to compare conditions. MetaTiME does have the functionality to output MeC values and top-most enriched cell state labels for each cell, which would facilitate cell counting based on independent component annotations. There are however several complications to this approach, requiring development of methodology that is somewhat separate from what we are describing. In particular, the independent component signatures do not necessarily define a partition of cells into clusters, as cells can be associated with more than one signature. For example, some cells in a sample might have the 'CD16 monocyte' signature as well as a 'macrophage IL1B' signature and a partitioning of cells would require a classification of cells based on both of these signatures. We have not investigated how best to do this analysis using MetaTiME signatures, although it is an excellent suggestion for further research.

Meanwhile, as the reviewer suggests, we have provided the function to count cell states after MetaTiME annotation in the toolkit. We provide example code to do quantitation of cell abundance depending on the cell metadata in which the user defines condition, sample, as well as the cell state label generated by MetaTiME annotator. Depending on the number of samples from a new dataset, the user can choose the testing statistics in comparing the cell abundance

or proportions. Thus, the MetaTiME github repo and the tutorial pages are updated to include differential signature analysis and the cell abundance analysis.

2. Most steps needed to reproduce the method are clearly explained but the GSEA analysis lacks details. It is not clear which exact method was used (incl. details such as the background used). I also wonder why only the first 100 genes were considered as GSEA naturally considers the ranks of all genes (in contrast to gene set overrepresentation analysis / hypergeometric test which is sometimes also called GSEA and should not be confused).

We thank the reviewer for this suggestion regarding the GSEA analysis. Our pathway analysis used gseapy and enrichr to obtain hypergeometric test-based enrichment from GO-BiologicalProcess, GO-MolecularFunction, Wikipathways, MSigDB-Hallmark, Reactome, Bioplanet, KEGG. We found it important to use top ranked genes like top 100 genes to compare to known markers, lineage transcription factors, ligand and receptors, to understand the MeCs, because most are about immunity that the above general pathway libraries are often not tailored for. Following the reviewer's suggestion, we have added an analysis that performs GSEA (gseGO function from ClusterProfiler,) using all genes ranked by weights from GO, KEGG, and Wikipathways. We have edited the text in lines 284, 465, added the corresponding GSEA columns in supplementary table 1, as well as the second sheet named "MeC_enrich". We have also edited Table S1 in the legend. We have included an extra citation, Yu et al, for this analysis.

3. To be honest I did not understand the part of the skewness of the scores. Skewness in one direction apparently informs about the directionality of the underlying function and the associated genes. At least I (probably others) would benefit from a supplementary figure where this is shown for a few selected examples.

We thank the reviewer for pointing out this source of confusion. In our analysis of independent components, we found that the distributions of weights were skewed in one direction or the other for each of the independent components. The genes in the direction with the longer tail tended to be more associated with various gene set annotations. Since the direction of independent component is arbitrary, we used scipy.stats.skewness to test whether skewness is on the positive or the negative side of the z-weight distribution. To clarify how we address the problem of finding the sign associated with functionality of independent components, we added one panel to Fig.S3. This figure illustrates skewness alignment through two examples of MeCs with different skewness signs, which is explained in the updated legend of Fig. S3. In the main text, we changed line 109 to refer to the new Fig. S3. We have also explained this in the Methods, line 436.

4. For future research it would be an interesting perspective to check if the components inferred in MetaTiME could be used as a source of signatures for the deconvolution of bulk RNA-seq data into more fine-grained cell types.

We thank the reviewer for this good suggestion about the potential application of MetaTiME on bulk RNA-seq data and will investigate this in the future development of MetaTiME.

Reviewer #3 (scRNAseq, systems biology) (Remarks to the Author):

Zhang et al. develop MetaTiME, a pan-cancer reference set of cellular identities and states for tumor microenvironment which can be used to annotate new datasets with continuous scores.

This is an area of active research and the general concept brought forward by the authors, while not entirely novel conceptually is enticing. The manuscript is clearly written and the authors provide well written, documented and working software implementation. However, I have a few concerns about the interpretation of the latent variables discovered by the authors across TMEs, and on how generalizable the method is for broad use by new users.

Below I provide a series of inquiries and recommendations to the authors that hopefully will help improve the manuscript:

We thank the reviewer for carefully reading our manuscript and for showing us ways in which it could be improved.

On the methodological approach:

1. The authors show that MetaTiME provides more resolved cell labels for an existing dataset which I assume is also used in the ICA training of the meta-components. One issue with that is that given enough time or effort, the authors of the original dataset could likely have generated a similarly granular annotation as well. In order to demonstrate the usefulness of MetaTiME in annotation of a broad set of tumor data, I propose the authors work with data that has not been seen for training. This could be accomplished by cross validation in a leave one cancer out strategy, in which all but one cancer type is used in training and then the left out cancer is annotated for the first time. The scores for TME cells in that held out cancer can then be compared with the scores in the fully trained model for evaluation, and this done for every cancer type. A second approach would be that the authors either generate data from a new cancer type or use a yet unused dataset from a novel cancer type for this purpose. I believe this would be important to demonstrate the usefulness of MetaTiME in annotating datasets in a much more real situation for new users.

We thank the reviewer for pointing out this important point. In our evaluation of MetaTiME we did not include the cells from Yost et al. as shown in Figure 3. We apologize for not explaining this crucial point more clearly in the main text. The Yost et al. data contain tumors of two cancer types, SCC and BCC. In the current MetaTiME training phase we used the cells from the SCC tumors and excluded cells from the BCC tumors. Thus, the “test cells” in Figure 3 are from BCC cells that are never seen by MetaTiME in the training steps. We have revised the text in line 195 to clarify the training-test separation: “We demonstrate the application of MetaTiME on basal cell carcinoma (BCC) single-cells from Yost et al. These test cells were excluded from the MetaTiME training stage.”

2. The authors claim batch effects are negligible by requiring MetaTiME meta-components to have samples from more than one batch. This is a positive step, albeit a relatively weak one as it does not guarantee that the sample structure in or across datasets is still not confounded by non-obvious sample groupings or even incorrect experimental designs in the original datasets. Could the authors explicitly compute the enrichment of different batches for each dataset in each meta-component, or complementarily do differential testing of meta-component scores between batches? This would reinforce that the discovered components reflect biology and not technical artifacts.

We thank the reviewer for this useful suggestion. The current design requires the minimum number of IC in a MeC to be 5, resulting in multi-cohort resources of each MeC. We have added columns in TableS1 to reflect MeC size and number of cohorts in each MeC. In response to the reviewer’s concern, we tested if MetaTiME meta-components could be driven by single datasets by leaving single datasets out of the meta-component calling. We have added a systematic “leave-one-out” test for all datasets to assess the robustness of the meta-components.

In particular, we added a supplementary figure for the leave-one-out test. As shown in Fig. S5, MeCs called from the full set and the MeCs called from leave-SCC-out have high correlation and high specificity. There is one parameter, the minimum number of ICs supporting a MeC, that can affect the included MeCs after leaving a single dataset out. When keeping the minimum number of ICs supporting a MeC as 5, there were two MeCs missing in the leave-SCC-out MeCs (Fig. S5a), these two MeCs being SCC dataset contributions. If we cut the minimum number of IC supporting a MeC to be 4, the two missing MeCs are rescued (Fig. S5b). In summary, the leave-one-out MeCs are highly correlated to the full-dataset MeCs based on the high correlation in both heatmaps (Fig. S5a,b). This is an advantage brought about by integrating a large number of datasets, which would not be achievable by standard ways of integrating cells, due to computing resource and time limits.

As suggested, we further performed leave-one-out tests for every dataset. We still used the same parameter of cluster calling resolution and the minimum IC number in one cluster to be 5. We used two metrics to check the similarity of each leave-one-out experiment compared to the full set. The first metric is mean maximum correlation between leave-one-out and the full set. This is calculated by taking the maximum correlation, row-wise and column-wise in the correlation heatmap, and then averaging. This metric reflects how strong the correlation is along the diagonal in the MeC correlation heatmaps. The second metric is the number of MeCs. As shown in Fig. S5c, the mean max correlation is high in each leave-one-out test (mean correlation=0.988). Similarly, the number of MeCs from each leave-one-out test is similar to the full set: mean MeC number across leave-one-out test is 84.64, close to the full set MeC number 86.

We think that data driven approaches will be affected by data size, and the batch effects will be better resolved as more data are incorporated. For future research, we would recalculate MetaTiME when the number of datasets significantly increases, as the tumor single-cell studies is still expanding.

In summary, we made the following improvement of the manuscript: we added one supplementary figure Fig. S5, edited the manuscript in line 115-120, and added one section in Method "MeC calling robustness with leave-dataset-out testing."

3. The authors provide no systematic validation of their interpretations of the meta-components, but however chose to give ample specific examples of genes enriched in signatures as justification for naming meta-components. While there is value in mentioning specific examples, the lack of a very systematic comparison of the proposed annotation for the metaclusters prevents the reader from assessing how good (or not so good) in general the provided interpretation is. It would therefore be great to assess how well the author's interpretations of the components' weight matches literature. One way could be for example to take the top N terms from each component and doing enrichment analysis with Enrichr for gene set libraries related with cell types, pathways, transcriptional regulators, and providing the relative enrichment in all of these as an additional tensor of information to be explored by the users of MetaTiME, and presented as validation of the manual annotation by the authors by observing how often each chosen term is recovered in the top enriched terms. In addition, it would be better if the nomenclature of the components were more systematic for example by always having first a broad cell type name, then a specific cellular state, as well as standardizing the capitalization and abbreviation (for example by not using abbreviations) of the meta-components.

We agree with the reviewer that a more systematic analysis of the components would be desirable, however, on investigation of the resources available for such a comparison we found no dataset that could be considered a gold standard. In naming and annotating the meta-components we tried to provide meaningful names and annotations, while also being consistent with accepted definitions of immunological terms. Acknowledging the limitations of our nomenclature we have therefore provided the significant enrichment terms from Enrichr and GSEA libraries in Table S1. In addition to pathway analyses used gseapy and enrichr to obtain hypergeometric test based enrichment from GO-BiologicalProcess, GO-MolecularFunction, Wikipathways, MSigDB-Hallmark, Reactome, Bioplane, KEGG, in the revision we have added an analysis that performs GSEA using all genes ranked by weights from GO, KEGG, and Wikipathways, and edited the text in line 296, 483, and added the corresponding GSEA columns in supplementary table 1, the second sheet named "MeC_enrich". We also edited Table S1 in the legend.

However, these pathway analyses do not always provide obvious and concise names that are suitable for interpreting and naming the MeCs as shown in the annotator. Thus, we try to give short names to the MeCs. In addition to the analysis based on gene sets, we also found it important to use the very top ranked genes to compare to known marker genes, lineage transcription factors, ligands and receptors, to understand the MeCs. In immunology, factors such as marker genes can sometimes provide useful indications of cell type that complement available gene signatures libraries.

To improve the naming, as suggested, we changed MeC names to follow a structure of "Category_Cell/State-Feature". Category marks 7 colored categories of MeCs, 6 related to lineages: B for "B cell", T for "T cells and NK cells", DC for "Dendritic cell", M for "Monocytes and macrophages", Myeloid for "Other Myeloid types besides monocytes and macrophages", Stroma for "Stromal cells", and the last category related to pan-cell type signaling pathway: Pan for "Pan-cell signaling pathway". Furthermore, we have also found that the previous several MeCs in the "Undefined" category is driven by the most highly ranked gene, with z-weight significantly (10-times) higher than the second gene in z-weight ranking. They represent pan-cell type features like ferritin activity in MeC-80 "Pan_Ferritin-FTH1", MeC-84 "Pan_Ferritin-FTL", ubiquitin-C activity in MeC-44 "Pan_Ubiquitin-C-UBC", steroid induction in MeC-79: "Pan_Steroid-induced-TSC22D3", actin in MeC-85 "Pan_Actin-ACTB". Thus, we annotated all of the 86 MeCs. The number of annotated MeCs are thus reflected in the manuscript.

The new naming is reflected in Table S1. Correspondingly, we changed the main figures: Fig1b, Fig2a,b,c,d, Fig3a,c,d, Fig4a,b,d, Fig5. ; the follow supplementary figures are updated: FigS4a,b, FigS7, FigS9 barplot, FigS10 text, FigS11d,c.

On data availability and software implementation:

- Thank you for making the software freely available to everyone.
- I commend the authors for the clearly documented software and availability of Jupyter notebooks exemplifying the usage of the software.
- Please make sure to make available the preprocessed datasets used in creating the TME reference as well in a publicly accessible database (e.g. Zenodo) to enhance reproducibility of the manuscript and reuse of resources across the community.

We agree that open-source runnable software would benefit the tumor single-cell field. As suggested, we have deposited processed data to Zenodo:
<https://doi.org/10.5281/zenodo.7410180>.

On the figures:

- Figure 3d is not mentioned.

We have edited the text on line 200 to refer to Figure 3d about marker gene expression in cell states.

REVIEWERS' COMMENTS

Reviewer #1 (Remarks to the Author):

The authors have added substantial additional analyses that were precisely what I wanted to see to confirm what was already apparent regarding the advantages of MetaTiME, but needed to be fully fleshed out. These new analysis are very thorough and directly address my primary critiques. All of my minor critiques were also directly addressed and I have no further criticisms of this work.

Reviewer #2 (Remarks to the Author):

I'd like to thank the authors for improving the manuscript and for addressing my questions and comments, especially for extending Figure S3 which now perfectly clarifies how the skewness measure is used. I have no further concerns.

Reviewer #3 (Remarks to the Author):

The manuscript has considerably improved and the authors have sufficiently addressed my concerns. I appreciate the effort the authors made in more systematically annotating the modules and making the information available.

I apologize for not recognizing the dataset in Fig 3 had not been part of the training.

Below are a few outstanding issues that regarding consistency:

- The methods section does include a section detailing GSEA analysis but not Enrichr using GSEApv.
- The files containing the weights and annotation of the MECs available at <https://github.com/yi-zhang/MetaTiME/tree/main/metatime/pretrained/mec> are tab-delimited but have a ".txt" ending. It would be more intuitive to users to rename them ending in ".tsv" (tab-separated values).

RESPONSE TO REVIEWER COMMENTS

Point by point responses to the reviewers' second round of comments are included below (reviewer comments are in blue, our responses are in black).

Reviewer #1 (Remarks to the Author):

The authors have added substantial additional analyses that were precisely what I wanted to see to confirm what was already apparent regarding the advantages of MetaTiME, but needed to be fully fleshed out. These new analysis are very thorough and directly address my primary critiques. All of my minor critiques were also directly addressed and I have no further criticisms of this work.

We thank the reviewer for the suggestions to improve our manuscript.

Reviewer #2 (Remarks to the Author):

I'd like to thank the authors for improving the manuscript and for addressing my questions and comments, especially for extending Figure S3 which now perfectly clarifies how the skewness measure is used. I have no further concerns.

We thank the reviewer for the suggestions to improve our manuscript and clarify concepts.

Reviewer #3 (Remarks to the Author):

The manuscript has considerably improved and the authors have sufficiently addressed my concerns.

I appreciate the effort the authors made in more systematically annotating the modules and making the information available.

I apologize for not recognizing the dataset in Fig 3 had not been part of the training.

Below are a few outstanding issues that regarding consistency:

- The methods section does include a section detailing GSEA analysis but not Enrichr using GSEAPy.

- The files containing the weights and annotation of the MECs available at <https://github.com/yi-zhang/MetaTiME/tree/main/metatime/pretrained/mec> are tab-delimited but have a "txt" ending. It would be more intuitive to users to rename them ending in "tsv" (tab-separated values).

We thank the reviewer for the suggestions to improve our manuscript and clarify the datasets usage.

We have edited the methods section to reflect Enrichr and GSEAPy. We have changed the file name in our repository and updated github.